# The Size-Dependent Photonic Characteristics of Colloidal-Quantum-Dot-Enhanced Micro-LEDs

**DOI:** 10.3390/mi14030589

**Published:** 2023-02-28

**Authors:** Kai-Ling Liang, Wei-Hung Kuo, Chien-Chung Lin, Yen-Hsiang Fang

**Affiliations:** 1Electronic and Optoelectronic System Research Laboratories, Industrial Technology Research Institute, Hsinchu 31057, Taiwan; 2Graduate Institute of Polymer Science and Engineering, National Taiwan University, Taipei 10617, Taiwan; 3Graduate Institute of Photonics and Optoelectronics, National Taiwan University, Taipei 10617, Taiwan

**Keywords:** quantum dots, micro-LED, micro-display, angular color

## Abstract

Colloidal CdSe/ZnS quantum dots (QD) enhanced micro-LEDs with sizes varying from 10 to 100 μm were fabricated and measured. The direct photolithography of quantum-dot-contained photoresists can place this color conversion layer on the top of an InGaN-based micro-LED and have a high throughput and semiconductor-grade precision. Both the uncoated and coated devices were characterized, and we determined that much higher brightness of a QD-enhanced micro-LED under the same current level was observed when compared to its AlGaInP counterpart. The color stability across the device sizes and injection currents were also examined. QD LEDs show low redshift of emission wavelength, which was recorded within 1 nm in some devices, with increasing current density from 1 to 300 A/cm^2^. On the other hand, the light conversion efficiency (LCE) of QD-enhanced micro-LEDs was detected to decrease under the high current density or when the device is small. The angular intensities of QD-enhanced micro-LEDs were measured and compared with blue devices. With the help of the black matrix and omnidirectional light emission of colloidal QD, we observed that the angular intensities of the red and blue colors are close to Lambertian distribution, which can lead to a low color shift in all angles. From our study, the QD-enhanced micro-LEDs can effectively increase the brightness, the color stability, and the angular color match, and thus play a promising role in future micro-display technology.

## 1. Introduction

Micro-LED has been regarded as a prospective display technology in the next generation because of its outstanding features of high scalability, high brightness, high contrast, fast response, and good stability [1,2]. In pursuit of high-quality display using micro-LEDs, the potential solution will provide high density pixels with vivid colors [3]. However, both of them are not easy targets to be achieved. Mass transfer and epitaxial growth for fabricating RGB micro-LEDs become extremely difficult as the chip size shrinks and the pixel density grows [4]. To make the situation worse, the external quantum efficiency (EQE) of the micro-LEDs tends to deteriorate as we reduce the size of the device [5,6,7,8]. Therefore, quantum dots (QD) color converting method, which can mitigate some of the aforementioned problems, has become one of the attractive options that can absorb blue light to generate color-transferred red and green light [9,10,11,12]. In addition to sustained quantum efficiency, the extra QD layer can provide an extra light source in the visible light communications [13,14], in which the modulated visible photons can be used for data transmission and provide an alternative solution of current Wi-Fi scheme.

The size-dependent EQE has been widely investigated [5,15,16]. In general, the InGaN micro-LEDs showed less inclination to decay when its size is reduced [7]. With its emission blue photons, the InGaN micro-LEDs is a natural choice as the color conversion light source [17]. On the other hand, blue shift of emission wavelength is detected with increasing injected current in InGaN micro-LEDs because of band filling effect, while red shift is observed in AlGaInP due to the self-heating effect [18]. The different behaviors of InGaN and AlGaInP red micro-LEDs can lead to undesirable change in the color gamut coverage which can directly affect the color quality of the display. The QD, on the other hand, is famous for their wavelength stability across various conditions [19].

In previous study [20], angular color shift problem of mixed RGB colors was detected in micro-LED displays due to the mismatched angular distribution between AlGaInP red and InGaN blue/green micro-LEDs. This is due to the different refractive indices of the two different material systems and the sidewall emission of the devices. The angular shift problems have been discussed in some micro-LED articles [21,22], but were seldom mentioned in QD on micro-LED structure. Hence, in this study, we will first place the QD composite on top of the device, which could have a stronger thermal influence on these nanocrystals. The size effect on the QD-enhanced micro-LEDs can also be observed via various devices with dimensions of 100 × 100, 50 × 50, 25 × 25 and 10 × 10 μm^2^. We then analyzed the angular distribution of blue and red QD micro-LEDs with size of 100 × 100, 50 × 50 and 25 × 25 μm^2^. We hope this photonic characterization can be helpful for the further integration of QD with the micro-LEDs.

## 2. Materials and Methods

A series of blue micro-LEDs and red QD micro-LEDs with size of 100 × 100, 50 × 50, 25 × 25, and 10 × 10 μm^2^ were fabricated from commercial 4-inch InGaN/GaN blue epitaxial wafers grown on sapphire substrates by a metal-organo chemical vapor deposition (MOCVD) system. For blue light micro-LEDs process, a layer of indium tin oxide (ITO) was deposited onto the wafer as the ohmic contact layer of p-type GaN. The mesa pixels of 100 × 100, 50 × 50, 25 × 25 and 10 × 10 μm^2^ were then defined by photolithography process, and followed by an etching process of inductively coupled plasms-reactive ion etch (ICP-RIE). After dry etching, a 100 nm dielectric layer of Si_3_N_4_ was deposited as a passivation layer by plasma enhanced chemical vapor deposition (PECVD). Then, the n-contacts and p-contacts were opened by ICP-RIE, and followed by Ti/Au layer was deposition with thickness of 100 nm. Next, a layer of black matrix (BM) was covered with 1 μm of black photoresist. The light-emitting areas were opened with the same size as micro-LEDs’ sizes by photolithography process. The n-contact pads and p-contact pads were also opened in this step as conductive electrodes. The finished blue light micro-LED structure is shown as Figure 1a.

The QD pixels were patterned directly on the top of the micro-LED mesas and were designed as the same size as the mesas, as shown in Figure 1b. The QD pixels were mainly composed of QD photoresist (QDPR), and QDPR was formulated by negative photoresist with 30 wt% of CdSe/ZnS QDs. The CdSe/ZnS QD powder was purchased from Unique Materials Co., Ltd. Starting from CdSe/ZnS QD powder, QDs were first dispersed in toluene solvent with 30 wt% (pristine QD). Pristine QDs were then transferred from toluene to propylene glycol methyl ether (PGMEA) by a rotary evaporator. Same weight of PGMEA was added into the QD-dispersed toluene solution. The mixed solution was then set on a rotary evaporator with a heating temperature of 60 °C to remove toluene. PGMEA was also the solvent of photoresist we used. The QD solution was then mixed with photoresist solution under vigorous stirring to form QDPR. Next, as a reference sample (called QD film), the QDPR solution was coated on a 4-inch glass substrate by a spin-coater with a spin rate of 800 rpm. The coated film was then soft baked at 100 °C for 3 min, and followed by an 80 Mj exposurein a SUSS aligner. QD pattern was formed as the same condition of QDPR film directly on the blue micro-LED wafer, but with a mask exposure and a further development of 0.05 wt% KOH aqueous solution for 60 s. In the last step, a layer of red color filter (CF) photoresist was patterned on the QDPR to reduce the blue light leakage. The structure of finished QD-enhanced micro-LEDs can been seen in Figure 1b.

The scanning electron microscopy (SEM) Images of blue and red micro-LEDs are shown in Figure 1a with a viewing angle of 20°. InGaN blue and QD-enhanced red micro-LEDs were located on the same wafer. Two top rows are blue micro-LEDs with BM opening, and the two rows at the bottom are QD red micro-LEDs with QD on the top of blue micro-LEDs. From left to right in columns are micro-LEDs in different sizes of 100 × 100, 50 × 50, 25 × 25 and 10 × 10 μm^2^. The chips were cut and bonded on a 3 × 3 cm^2^ printed circuit board (PCB), shown as Figure 1b, for further photoelectric measurement. For electrical characteristic measurement, we used a probe station connected to an IV2400 network analyzer. For external quantum efficiency (EQE) analysis, the micro-LEDs were measured in an integrated sphere coated with the BaSO_4_ material, and Keithley 2401 source meter were used to supply currents to the micro-LEDs under test. The brightness and spectral data of LEDs were measured by a 2D spectrometer from TOPCON Company. Keithley 2401 source meter are also used to provide different currents for spectra investigation. For far-field patterns analysis, the micro-LEDs on PCB samples were fixed on a rotary stage, which is perpendicular under the 2D spectrometer. The samples of 100 μm, 50 μm, and 25 μm in both blue and red LEDs were rotated and measured from +80° to −80° and under the same current density of 400 A/cm^2^.

## 3. Results and Discussion

Figure 2 shows the typical forward- and reverse-biased J-V curves of our blue micro-LEDs with different sizes. In general, the devices exhibit similar trend as reported previously [15,23]. Due to difficulties in smaller area in p-type contact, we would expect lower currents at the same forward voltage for small devices. The leakage current density for the small devices (such as 5 μm ones) is still higher, indicating the larger portion of carrier might be consumed in the sidewall. The reverse current of 100 × 100, 50 × 50, 25 × 25 and 10 × 10 μm^2^ single pixel at -5 V are 4.12 × 10^−7^ A, 3.16 × 10^−7^ A, 2.75 × 10^−7^ A and 1.99 × 10^−7^ A, respectively.

Figure 3a shows the EQE of different micro-LED sizes as a function of current density. The maximum EQE tends to drop with reducing micro-LED sizes. The maximum EQE of micro-LED size of 100, 50, 25 and 10 μm are 20.0%, 19.8%, 18.1%, and 13.2% respectively. In addition, the current density at maximum EQE increased with decreasing micro-LED size. This phenomenon could be attributed to higher non-radiative recombination through sidewall defects in smaller LED size [13]. In Figure 3b, high luminance with over 10^6^ cd/m^2^ was achieved at current density greater than 130 A/cm^2^. In addition, higher brightness was observed in the larger size of blue light micro-LED of 100 and 50 μm. The luminance of 100, 50, 25 and 10 μm blue micro-LED at 130 A/cm^2^ are 1.4 × 10^6^ cd/m^2^, 1.2 × 10^6^ cd/m^2^, 1.0 × 10^6^ cd/m^2^ and 1.1 × 10^6^ cd/m^2^, respectively.

The measured *EQE* can be analyzed by the modified ABC model previously proposed as Equation (1) [5]:

External quantum efficiency (*EQE*):(1)$$EQE=\frac{\eta_{LEE}\left(1-\beta n\right)Bn^{2}}{\left(An+Bn^{2}+Cn^{3}\right)}.$$
where $\eta_{LEE}$ is the light extraction efficiency of the micro-LEDs device, *β* is the leakage ratio of the device, *A* is the Shockley–Reed–Hall (SRH) coefficient, *B* is the bimolecular coefficient for radiative recombination, and *C* is the Auger recombination coefficient. The magnitude of coefficient *A* can be a good indicator of the non-radiative recombination within the device. The light extraction efficiency is close to constant when different current levels are applied to the diode. The numerical fitting can be accomplished by comparison between the measured and calculated results. As shown in Figure 4, our numerical calculation is close to what were measured. The maximum of *EQE* for devices of different sizes is decreased from 20% (100 μm) to 13.2% (10 μm), while the current density at this EQE_max_ also shifts from 8 A/cm^2^ (100 μm) to 20 A/cm^2^ (10 μm). The extract SRH coefficients (*A*) and the leakage coefficient (*β*) are also listed in Table 1. As we expected, the SRH coefficient rises when the size of the device reduces due to the increased sidewall to the volume ratio and thus the increased non-radiative recombination in the sidewall traps. These findings are in line with what we reported before.

The electroluminescence (EL) spectra under different injected currents at a 10 × 10 μm^2^ blue micro-LED are shown in Figure 5a, and the peak wavelength shifting of all sizes blue micro-LED are shown in Figure 5b. The EL intensity raised with increasing current density in all sizes, and the spectra blue-shifted with increasing current density. The peak wavelength of 100 μm blue micro-LED appears a 6 nm blue shift as current density raised from 1 to 300 A/cm^2^, and 10 μm blue micro-LED, on the other hand, shows a blue shift of 5.7 nm. The blue shift would be attributed to the band filling effect and quantum-confined Stark effect [16,24]. The changes in photonic characteristics of blue micro-LEDs with different sizes are listed in Table 2.

After the characterization of blue micro-LEDs is finished, we can test the QD-coated ones. The first property to be noted is the emission spectrum of these devices before and after the coating procedure. Figure 6 shows the PL of the pristine QD, QDPR film, and patterned QDPR with color filter (CF) on the micro-LEDs. From the measured spectra, one can define the percentage of QD photons in the overall emission by the light conversion efficiency (LCE) [25]. Its expression can be found in Equation (2). 

Light conversion efficiency (LCE):


(2)
$$\mathrm{LCE}=\frac{\#\;of\;QD\;emitted\;photons\;}{\#\;of\;total\;detected\;photons}=\frac{\int_{QD\_band}^{}\frac{\lambda }{hc}\times I_{em}^{QD}\left(\lambda \right)d\lambda }{\int_{total}^{}\frac{\lambda }{hc}\times I_{em}^{}\left(\lambda \right)d\lambda }.$$


The LCE is a good parameter to evaluate the purity of the detected color of a QD-coated micro-LED. In the ideal case, the LCE is 100%, meaning no leaked blue photons. In our case, the LCE of the QDPR film is around 80% while this number drops to 75% when QD is coated on the device top. The cause of this decrease in LCE is due to the reduction in the QDPR thickness on the micro-LEDs. Even though we used the same process parameters, the QDPR thickness drops to 2.3 μm from its thin film case of 3.4 μm. The complex morphology of the processed wafer can lead to an uneven distribution of the QDPR, and the viscous solution of QDPR tends to accumulate at the bottom of the trench and thus less QDPR can be coated on the top surface [26]. Major difference in remaining blue photons can be seen between the QD film and the QD pixel cases. It is noted that the CF we applied here only blocked the wavelength shorter than 580 nm as shown in Appendix A. Therefore, the CF did not cause the shift of red light’s peak wavelength, nor the change in FWHM in this study. If we normalized the spectra for all three cases as shown in Figure 6, one would observe that the shape and the peak position of the QD emission are actually quite close. The detail optical characteristics and thickness of these three material cases are also listed in Table 3.

Figure 7 shows the devices under current injection, and as expected, the QD-enhanced red micro-LEDs showed purplish color indicating residual blue photons in the QD red pixels. The overall luminance, including the blue and red photons, vs. current density of the QD-enhanced red micro-LEDs of different sizes are graphed in logarithmic scale in Figure 8. The output light looks saturated at high current, and for larger devices (such as 100 and 50 μm ones), the high luminance of 1.9 × 10^6^ cd/m^2^ and 1.4 × 10^6^ cd/m^2^ can be achieved. Meanwhile, the smallest device (the 10 μm one) has a much lower luminance at the level of 6.0 × 10^3^ cd/m^2^, which is more than two orders of reduction. The output power of InGaN red micro-LED decreased as the chip size shrunk from 100 μm to 10 μm, but the intensity change was within an order (5000–2500 mW/cm^2^) [16]. In the traditional AlGaInP devices, as the size shrinks, the output power and quantum efficiency decrease dramatically [5,17,27,28]. As shown in previous paper, the brightness can drop almost several hundred times when the device becomes 10 times smaller. On the other hand, our QD micro-LEDs can maintain their luminance level better under such device scaling. Compare to small size AlGaInP red micro-LEDs, QD micro-LED still appeared higher brightness. A 16 μm AlGaInP red micro-LED showed brightness between 10^0^ and 10^1^ cd/m^2^ [27], while our 10 μm QD micro-LED attained 10^3^–10^4^ cd/m^2^ at 50 A/cm^2^ in this study.

As shown in Figure 9, if we used the 100 μm device as the standard to normalization, we could evaluate the red color emission at 50 A/cm^2^ of our QD-coated micro-LED with different sizes and compare them with previous publications. The distinctive stable red emission of our QD-LED larger than 25 μm can be observed. The normalized red emission intensity of 10 μm devices drops significantly to less than 1% in our case. However, this is still better than those of AlGaInP-based devices. The cause of this significant reduction is still under investigation, and it could be attributed to the reduction in the QDPR thickness as well.

EL spectra of 10 × 10 μm^2^ of red QD micro-LEDs are shown in Figure 10a. The EL intensity is increased with rising current density, but the peak wavelength shifting is relatively small. We fitted the red-light spectra by Gaussian function and calculated the peak wavelength difference. The results were plot against the current density as shown in Figure 10b. The changes in photonic characteristics are also listed in Table 2. The differences in peak wavelength position are small in all sizes. For instance, 100 μm QD LED shows a 1.7 nm red shift (Δλ = 1.7 nm) as current density increases from 1 to 300 A/cm^2^, and 10 μm shows a shifting of less than 0.1 nm. Compared to previous study, in which both of the InGaN and AlGaInP devices were tested and showed much greater wavelength shift (Δλ > 50 nm for InGaN red micro-LEDs [18], and Δλ = 2.7 nm for AlGaInP red micro-LEDs [28]), our QD-enhanced red micro-LEDs exhibit a better spectral stability. The blue shift of InGaN red micro-LED is caused by QCSE and band filling effects [18]. This phenomenon was also detected in our blue micro-LEDs. As shown in Table 2, the peak wavelength of 100 μm blue micro-LED had a blue shift of 5.9 nm as the current density increased from 1 to 300 A/cm^2^. However, after QD coated, the peak wavelength shifting constricted down to only 1.7 nm. The emission wavelength of QD is determined mostly by the composition material and the size of nanoparticles [29,30,31]. The injection current of LED device should not affect the emission wavelength of the QDPR layer in the first order. However, the direct contact between the QDPR and the pumping micro-LED might lead to thermal conduction form the LED to the QDPR layer. While the emission peak of a semiconductor-based device can blue shift several tens of nanometers [18,32], we can see almost no change in the low current range when we check the QD-emitted peak shift in the red band and a slight red shift (less than 2 nm) in large device due to higher junction temperature induced at high current.

Although QD micro-LED exhibit high brightness and stable spectral characteristic, there is still an issue that can be observed from the spectra. The blue light leakage at 445–450 nm shown in Figure 11 indicates the concentration of QD, even combined with the CF layer, was not high enough to absorb all the blue light emission from the micro-LEDs below. Figure 11a shows LCE of different micro-LED sizes at 300 A/cm^2^ and 1 A/cm^2^ current densities. In the LED size of 100, 50, and 25 μm, LCE are higher with lower current density of 1 A/cm^2^. This phenomenon can also be observed in Figure 11b: LCE decreased with increasing injected current density at size of 100, 50, and 25 μm. This indicates that blue light output increases higher than red light as the current density rises. However, the 10 μm QD micro-LED shows an opposite result with CCE increased from 71.0% to 80.2% as the current density increased from 1 A/cm^2^ to 300 A/cm^2^. The issue of blue light leakage can be further overcome with a high optical density color filter (CF) layer or a distributed Bragg reflector (DBR) layer on the top of the QD layer [33].

The last topic we investigate is the device’s angular emission pattern. In a display, the color difference in different viewing angles can cause a serious color shift and deteriorate our viewing experience. Unfortunately, due to differences in materials, index of refraction, and chip thickness, we observed a very different angular pattern from individual RGB micro-LEDs in the past [21]. To check this property in our devices, the micro-LEDs were mounted on a rotational stage to scan through various angles, as described in previous section. Figure 12 shows the far-field radiation pattern of blue and red QD micro-LEDs with 100 × 100, 50 × 50, and 25 × 25 μm^2^ at the same current density of 300 A/cm^2^. The dots present the measured data and the black dashed line stands for Lambert’s cosine law. The red-light radiation is slightly wider than Lambertian distribution at the angle of 50°, which could be explained by extra QDPR protruding out of the BM plane. Meanwhile, the BM can effectively modify emission patterns of all three colors into a Lambertian-like angular distribution among various sizes of pixels. Compared to a real RGB micro-LED panel, the sidewall emission from a bare micro-LED without black matrix can affect the far-field pattern easily and lead to the aforementioned color shift issue. The light emission from sidewall gradually increases especially as the chip size down to micrometer scale. The resulting twin-peak distribution is often observed in previous mini- or micro-LEDs [20,21]. However, in this study, our light sources (or pixels) can be treated more as a planar light source due to the BM and the isotropic light from QD conversion light. The angular distribution match between blue light micro-LEDs and QD-converted micro-LEDs can avoid the color shift at a large viewing angle and provide a practical application in wide view display. Moreover, angular distribution of QD micro-LEDs panel is less size-dependent as shown in Appendix A. The result confirms that the QD/BM converted structure provides a symmetric and size-independent angular distribution. Therefore, the quantum-dot-enhanced micro-LED display can provide a stable mixing color at all polar angles.

## 4. Conclusions

The size effect and angular emission pattern of blue light and QD color-converted red micro-LEDs were discussed in this study. We demonstrated the blue and red micro-LEDs with size of 100 × 100, 50 × 50, 25 × 25 and 10 × 10 μm^2^ on a 4″ InGaN/GaN wafer. Blue light micro-LEDs, as well as the back light sources of red light QD micro-LEDs, show the electrical and optical characteristics of low current leakage of 10^−7^ A at −5 V, maximum EQE of 20.0–13.2%, and brightness over 10^6^ cd/m^2^ at 130 A/cm^2^. The EL spectra of blue micro-LEDs blue shifted ~6 nm with increasing injected current density due to screening of QCSE and band filling effects. However, red light micro-LEDs converted by QD show stable EL spectral characteristics. The peaks shifted comparably small as the current density increased, and were also size-independent. This behavior is very different from that of AlGaInP or InGaN red micro-LEDs. However, the blue light leakage from the EL spectra were still observed. The LCE of 71–86% indicated that QD micro-LEDs still need further improvement by higher optical density of CF or a DBR layer to increase the light color purity. The angular emission pattern of blue and QD red micro-LEDs were measured and shown as Lambertian-like distribution with the black matrix on the top. The similar angular distribution shows that QD converted structure could provide a stable mixing color in a large viewing angle. With the stable EL spectral characteristics and the homogeneous light distribution, we hope that this study shows QD conversion as a prospective technology for full-color micro-LED display in the near future.

## Data Availability

Not applicable.

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
