# Peer review of "The Size-Dependent Photonic Characteristics of Colloidal-Quantum-Dot-Enhanced Micro-LEDs"

_micromachines, 2023, doi:10.3390/mi14030589_

Round 1

Reviewer 1 Report

Dear Editor,

I received the manuscript “The Size-dependent Photonic Characteristics of Colloidal-Quantum-Dot-Enhanced micro-LEDs” for review.

The manuscript reports on the fabrication of micro-LEDs using the direct photolithography method. The effect of  CdSe/ZnS quantum dots’ presence on the top of micro-LEDs was studied and described in detail. The manuscript is well-written from a scientific point of view.

I have only a few suggestions for improving this manuscript:

-           -       The first sentence in the abstract: Colloidal quantum dot enhanced micro LEDs with size varying from 10 to 100 micrometers were fabricated and measured. In my opinion, this sentence should be corrected. For example, “Colloidal CdSe/ZnS quantum dots enhanced micro-LEDs with sizes varying from 10 to 100 micrometers were fabricated and measured”. I would like to draw your attention to the abbreviations "QD" and "QDs" of quantum dots - in the manuscript I have found both. Please note, that in the text should be only one of them. Also, you should check the entire text to correct the phrase “micro LEDs”, it should be with a dash.

-           -    The text (positions 77-79) is unclear (two sentences). Please reformulate it.

-          -    How was the synthesis of QDs performed? How were the pristine QDs transferred into propylene glycol methyl ether? Please give a detailed description.

In my opinion, this manuscript should be accepted for publication after minor revision.

Best regards

Reviewer 2 Report

In this manuscript, the authors reported the size effect and angular emission pattern of blue light and quantum-dot color-converted red micro-LEDs. The red light micro-LEDs converted by quantum dots show stable EL spectral characteristics different from AlGaInP or InGaN red micro-LEDs. I would recommend the acceptance of this manuscript after major revision. There are a few queries which must be addressed.

1.     The introduction part should be improved it is owing to the lack of novelty and significant achievements in the current form. Some references are relevant to this topic, which could help readers better understand the latest progress: Nanomaterials, 2022, 12(4): 627.; ACS applied materials & interfaces, 2018, 10(6): 5641-5648.

2.     The mechanism behind stable EL spectral characteristics of red-light micro-LEDs converted by QDs different from AlGaInP or InGaN red micro-LEDs is poorly explained, which is the major drawback of this work. It must be properly addressed.

3.     The use of “quantum dots” and “QDs” should be uniform in the full text.

4.     Some errors should be avoided, such as “The red light QD Then the QD-enhanced red micro-LED 90 was completed and shown as Scheme 1b.”、“Figure 5b show the peak wavelength shifting of all sizes..
